# Influence of Activators on Mechanical Properties of Modified Fly Ash Based Geopolymer Mortars

**DOI:** 10.3390/ma13051033

**Published:** 2020-02-25

**Authors:** Piotr Prochon, Zengfeng Zhao, Luc Courard, Tomasz Piotrowski, Frédéric Michel, Andrzej Garbacz

**Affiliations:** 1Department of Building Materials Engineering, Faculty of Civil Engineering, Warsaw University of Technology, Armii Ludowej 16, 00-637 Warsaw, Poland; 2Urban and Environmental Engineering, Université de Liège, Allée de la découverte 9, Sart Tilman Campus 4000 Liège, Belgium

**Keywords:** geopolymers, alkali-activated materials, mortars, co-combustion fly ash, biomass, alkaline activation, mechanical properties

## Abstract

The aim of this work was to study the influence of the type of activator on the formulation of modified fly ash based geopolymer mortars. Geopolymer and alkali-activated materials (AAM) were made from fly ashes derived from coal and biomass combustion in thermal power plants. Basic activators (NaOH, CaO, and Na_2_SiO_3_) were mixed with fly ashes in order to develop binding properties other than those resulting from the use of Portland cement. The results showed that the mortars with 5 mol/dm^3^ of NaOH and 100 g of Na_2_SiO_3_ (N5-S22) gave a greater compressive strength than other mixes. The compressive strengths of analyzed fly ash mortars with activators N5-S22 and N5-C10 (5 mol/dm^3^ NaOH and 10% CaO) varied from 14.3 MPa to 5.9 MPa. The better properties of alkali-activated mortars with regular fly ash were influenced by a larger amount of amorphous silica and alumina phases. Scanning electron microscopy and calorimetry analysis provided a better understanding of the observed mechanisms.

## 1. Introduction

Ordinary Portland Cement (OPC), within its nearly two centuries of history, become an essential component of the built environment. During the last decade, its production increased significantly from 2.9bn tons in 2008 to over 4.65bn tons in 2016 [1,2]. The production of OPC is widely recognized as one of the major contributors of greenhouse gas emissions by emitting, depending on the source, from 6–8% of all anthropogenic-based carbon dioxide into the atmosphere [3,4,5]. The elevated carbon footprint is primarily associated with its industrial high fuel requirements (mostly satisfied by fossil fuels) and to the chemical production process, which releases a large amount of CO_2_ as a byproduct [2,6]. With increased awareness in the European Union of the greenhouse gases emission from the energy production and industrial processes, some mitigation strategies include diverse approaches such as consideration of an increase of the share of renewables in energy production, use of biomass fuels, an increase equipment efficiency, preparation of environmentally-friendly concrete composition, use of alternative raw materials, and use of material substitutes [7,8].

One of the possible paths for mitigating the CO_2_ emissions from OPC production and the problems associated with it is the development of a new type of alternative, sustainable material obtained from industrial wastes including the geopolymer [9,10]. Geopolymer composites are reported to be much more environmentally-friendly than OPC in terms of lower energy requirement for production with significantly less CO_2_ emissions [11]. Those composites are comparable with OPC materials since they present good compressive strength, excellent chemical resistance, and minor drying shrinkage [12,13]. The name “geopolymers” was first introduced by J. Davidovits to describe mineral polymers received from geosynthesis or geochemistry [14]. Those amorphous polymers are formed by mixing an aluminosilicate raw material with aqueous alkali solution [15]. To obtain geopolymer composites with specific properties, it is important to select adequate raw material, mix it with alkaline solution of appropriate molar concentration and Si/Al ratio, and cure it either at an ambient temperature or an elevated temperature [16,17,18]. As a binder metakaolin [19,20], fly ashes [21,22,23] or slags [24,25] are the most commonly used pozzolanic materials.

The geo-polymerization needs an alkaline activator, among which the most popular are sodium/potassium silicate and sodium/potassium hydroxide (NaOH, KOH) [26,27,28]. The type of used activator has a high influence on the raw materials dissolution of Al and Si ions. Jaarsveld and Deventer pointed out that using NaOH solutions in geopolymers’ binding process gives a higher degree of condensation, compressive strength, lower degree of crystallinity, and acid resistance than those prepared with KOH solutions [29]. Using NaOH with water glass, which increases the Si content in geopolymer mix, provides improvement in mechanical properties of the composite [30]. The most common molar concentration of NaOH used in research is not higher than 20M and, in accordance with research [31], sodium silicate solution to NaOH content should be not more than 1.0.

In FA-based geopolymers, the geopolymerization mechanism analysis is complicated by the presence of crystalline phases: mullite, haematite, or quartz [32]. In accordance with Fernandez-Jimenez et al.’s research, the FA composition is a major factor for proper geopolymers binding process. The minimum level of reactive silica and alumina must be obtained in the FA compound to guarantee a set off of the binding reaction [33,34]. In case of fly ashes characterized by higher levels of calcium compounds, it is suggested to use the specific acronym “alkali-activated materials (AAM)” for the binders rather than term geopolymers [35]. Geopolymers usually require a high temperature for activation and solidification while AAM can be used at ambient conditions. While in AAM, the main hydration product is calcium-(alumino)-silicate-hydrate (C-(A)-S-H) in geopolymers. C-(A)-S-H occurs alongside with sodium-alumino-silicate-hydrate (N-A-S-H) [35,36]. During every year, more coal power plants convert to co-firing or open new power blocks operational with biomass combustion. It is fundamental to analyze and recognize the suitability and quality of this changed in combustion process fly ashes for their possible utilization as geopolymer binder [37,38] or AAM [36].

In this study, three fly ashes of different origin and chemical characteristics were examined as possible precursors for alkali activation with different activators: silica fly ash from coal combustion (RFA), co-combustion fly ash derived from combustion of coal and wooden biomass (CFA), and fly ash derived from biomass combustion (BFA). All fly ashes were examined with an X-ray Fluorescence Test (XRF) and an X-ray Diffraction Test (XRD) as well as SEM image analysis with energy dispersive X-ray spectrometer (EDS) attachment along with density, loss on ignition, and particle size distribution measurements.

To assess if biomass ashes and co-fired fly ashes can be viably used as alkali-activated materials, analyzed fly ashes were alkali-activated with different activators: (a) sodium hydroxide (N5, N10), (b) quicklime (C10, C15), (c) mix of sodium hydroxide and quicklime (N5-C10), or (d) mixes with sodium silicate, (N5-S22, C10-S22). Obtained pastes and mortars were analyzed with SEM/EDS analysis, thermogravimetric analysis (TGA), and calorimetry. Understanding the influence of different alkali activators on fly ashes derived from co-combustion of coal and biomass or combustion of biomass can be valuable for analyzing their potential reuse as precursors for AAM and their possible implementation as building composites.

## 2. Materials and Methods

### 2.1. Fly Ashes

Three different fly ash (FA) samples were alkali-activated (see Table 1). The first one is coming from bituminous coal combustion (RFA). The second one is derived from co-firing (co-combustion of coal and wooden biomass - CFA) and the last one was obtained from agricultural biomass combustion (BFA). Both RFA and CFA were obtained from the Polish Power Plant. The examined CFA was produced by full-scale co-firing trials in which bituminous coal was fired with a wooden biomass in not more than 30% weight percentage replacement of coal with the same processing conditions as for the RFA. The BFA produced from agricultural waste combustion was received from an industrial partner of GeMMe Department at the University of Liege (Liege, Belgium).

#### 2.1.1. Density

The real density of fly ashes was measured by a gas pycnometer. This apparatus operates by detecting the pressure change resulting from displacement of gas by a solid object between two chambers including the sample chamber and the referenced chamber.

A pulverized sample of fly ash of unknown volume (V_m_) and known mass (m_m_) was placed into a sealed sample chamber of known volume (V_ch_). After sealing, the pressure within the sample chamber was measured (P_ch_). The isolated reference chamber of known volume (V_r_) was charged with a greater pressure than P_ch_ (P_r_). Opening the value isolating the two chambers allowed us to equilibrate the system pressure (P_sys_). The test was repeated three times for one mortar. The volume of the sample was determined in line with the gas law from the equation using the mean results.
(1)$$V_{m}=\frac{\left(P_{sys}\times V_{ch}+P_{sys}\times V_{r}-P_{ch}\times V_{ch}-P_{r}\times V_{r}\right)}{\left(P_{sys}-P_{ch}\right)},\;\left[dm^{3}\right]$$
where:
V_m_—volume of pulverized sample, dm^3^m_m_—mass of pulverized sample, gV_ch_—volume of sealed chamber, dm^3^P_ch_—the pressure within the sample chamber, PaV_r_—reference chamber of known volume, dm^3^P_r_—charged pressure, PaP_sys_—system pressure, Pa

The density of fly ash samples was calculated from the density definition.
(2)$$\rho =\frac{m_{m}}{V_{m}},\;\left[\frac{g}{dm^{3}}\right]$$

#### 2.1.2. XRF and XRD

The oxide composition, mineral phase characterization and amorphous content of the fly ashes presented in Table 2 was determined using X-ray Fluorescence Test (XRF, Thermo Fisher Scientific, Waltham, MA, USA) and X-ray diffraction (XRD, Bruker, Billerica, MA, USA). XRF samples were fused with lithium borate at a temperature over 1000 °C and the resulting beads were analyzed by X-ray fluorescence spectrometry. For XRD, FA samples were dried, and sieved though a 63-µm sieve. To obtain quantitative data, a known volume of corundum was added to the FA powder. The obtained sample was grounded and put into an empty sample holder. The sample was gently pressed with a glass slide. The excess powder from the sample holder edges was removed and it was carefully placed in the appropriate XRD slot. The XRD diffractograms of fly ashes are presented in Figure 1.

#### 2.1.3. Loss on Ignition (LOI)

Loss on ignition (LOI) of fly ashes was determined gravimetrically in accordance with EN 196-2 and EN 450-1. 1 ± 0.05 g of fly ash was put into a crucible (m_1_). The covered crucible was placed in the furnace at 950 ± 25 °C. After heating for 5 min, the lid was removed from the crucible and the crucible was left in the furnace for the next 55 min. After that, the crucible cooled down to room temperature in a desiccator. Constant mass (m_2_) was determined by making successive 1-h ignitions, which were followed each time by cooling and weighting. The LOI was calculated as:(3)$$\mathrm{LOI}=\frac{m_{1}-m_{2}}{m_{1}}\times 100\;\left[\%\right]$$

#### 2.1.4. Particle Size Distributions

The laser diffraction testing instrument Malvern Mastersizer 2000 (Malvern Instruments, Malvern, UK) with dispersion unit Hydro 2000 S (Malvern Instruments, Malvern, UK) was used for obtaining FA particle size analysis shown in Figure 2. Particle size distributions (PSDs) of fly ashes were measured by a wet dispersion laser diffraction method (particles diameter range detection 0.02 µm to 2000 µm). This equipment uses the Mie theory and principles of static light scattering to calculate the size of particles in a sample. Basically, small particles should scatter light at large angles when large particles should scatter light at smaller angles. The scattering pattern produced by the powder sample is recorded and, by using the Mie scattering theory, the particle size distribution can be calculated.

#### 2.1.5. SEM Image Analysis

Thermal Field Emission Scanning Electron Microscope (FE-SEM, Hitachi, Tokyo, Japan) accompanied by a Bruker EDS analyzer was used for morphology analysis of fly ashes and mortar samples. Fly ash samples were applied on a carbon tape to create a thin layer of powder. After that, they were coated with a thin gold film. The images were taken at varying magnifications under high vacuum and accelerating voltage of 5 kV. The mortar specimens were taken from beam samples 4 × 4 × 16 cm. A 1-cm thick samples were cut from the beams, dried, and prepared for the SEM by covering with a thin gold film in vacuum conditions.

### 2.2. Alkali Activated Mortars

The study was performed on alkali-activated materials (AAM) designed according to the following principles.
−Aluminosilicate precursors: fly ashes;−Alkaline activators: sodium hydroxide, sodium silicate, and quicklime;−Fine aggregate: sand.All alkali-activated mortar mixes were prepared and manufactured according to a modified procedure based on EN 196-1 as follows:−Adding aluminosilicate precursor to the mixer pan and activating the mixer at low RPM (rotational movement - 140 ± 5 RPM-1).−Pouring the sand in at an even rate for the first 30 s of mixing.−Adding alkaline activator solution at an even rate for the next 30 s of mixing (the “zero time” for setting time measurement).−Switching the mixer to high RPM (rotational movement 285 ± 10 RPM-1) and continuing mixing for 30 s more.−Stopping the mixer after a total of 90 s and following the EN 196-1 standard procedure for preparation of mortars.

Samples were kept in laboratory conditions for the first 4 h and then were cured in a dryer at a temperature of 65 °C for the next 4 h. After the heat curing, all samples were kept in laboratory conditions (20 ± 2 °C and 60% relative humidity) until testing. Samples were demoulded after 24 h and kept in laboratory conditions described above.

Fly ashes (RFA, CFA, or BFA) were mixed in accordance with the above procedure with seven different types of alkaline activators. The aqueous hydroxide solutions were prepared with sodium hydroxide pellets. Sodium silicate was added in liquid form. The specimen names in Table 1 based on general notation “XXX-Y#-Z#”, where “XXX” is the type of FA (RFA, CFA, BFA), “Y#” and “Z#” are symbols of alkali-activators (N5, N10 for 5M, and 10M sodium hydroxide, C10 and C15 for 10% and 15% substitution of FA by quicklime, S22 for 100 g of sodium silicate). Flexural and compression tests, Thermogravimetric analysis (TGA, Navas Instruments, Conway, NH, USA), Scanning Electron Microscopy (SEM), and Calorimetry were used to examine properties of obtained alkali-activated materials.

#### 2.2.1. Compressive and Flexural Strengths

The compressive and flexural strength tests were performed according to the standard PN-EN 196-1 using beam samples 4 × 4 × 16 cm and a hydraulic press with appropriate attachments. Tests were conducted after 7 and 28 days of mortar preparation. The tests were performed on controls and an Instron hydraulic press (Instron, Norwood, MA, USA) with a loading rate of 0.33 MPa/min and sensitivity of 100 kN for the compressive strength and a loading rate of 0.017 MPa/min with sensitivity of 5 kN for flexural strength. The result was the average of three samples for the flexural strength test and of six samples for compression strength.

#### 2.2.2. TGA

Thermogravimetric analysis of the mortars was performed up to 1000 °C with a heating rate of 10 °C/min. The tested mortars were crushed and then pulverized into a fine powder for analysis. All samples were held at 105 °C for about 30 min to release any free water before subjecting them to a higher temperature (m_105_). Weight loss was measured as a function of temperature.

The analyzed bound water of the hydrate phases (H) and the Ca(OH)_2_ content (CH) were calculated from the weight loss curve expressed as a percentage of the dry sample weight at 1000 °C (m_1000_) using Equations (4) and (5) based on literature [39,40,41,42]. The CaCO_3_ content was determined similarly with Equation (6).
(4)$$H=\frac{m_{105}-m_{550}}{m_{1000}}\times 100\%$$
(5)$$CH=\frac{m_{410}-m_{550}}{m_{1000}}\times 100\%$$
(6)$$CC=\frac{m_{550}-m_{800}}{m_{1000}}\times 100\%$$
where m_n_ is the dry sample weight at the n temperature, °C.

#### 2.2.3. Calorimetry

The calorimetry measurement was simplified to a measurement of temperature change in a function of time of selected geopolymer pastes. Dry material was mixed with the alkaline activator in plastic containers placed in insolated polystyrene boxes with installed thermocouples. The alkali-activation process was tested in laboratory conditions (20 ± 2 °C and 60% relative humidity) for 24 h.

## 3. Results and Discussion

### 3.1. Fly Ashes

#### 3.1.1. The Chemical Compositions, Mineral Phase Characterizations, and Physical Characteristics

The chemical composition and physical characteristic of the fly ashes is presented in Table 2. RFA exhibits the chemical properties required in EN 450-1 and in ASTM C618 corresponding to Class F fly ash (approximately 83.07% of the RFA composed of silica, aluminum, and iron oxides). The high level of Si and Al components is important in the geopolymerization process mainly because of their contribution to the strength development that occurs due to alkaline activation [43,44]. Low content of calcium oxide (2.70%) and average loss of ignition (LOI = 5.51%) comply with the specifications defined in the standards mentioned above. Predominant crystalline phases presented in the RFA XRD diffractogram (Figure 1) were mullite and quartz, as mentioned by Vickers et al. [32] or Jimenez et al. [33].

The silica, aluminum, and iron contents for CFA are lower compared to the RFA samples (SiO_2_ + Al_2_O_3_ +Fe_2_O_3_ = 54.47%). The decrease was also observed in amorphous content (53.13%). It is possible that these compositions differ because of the biomass fraction added in combustion (wooden biomass fly ashes are primarily crystalline) as it was observed in other research studies [45,46]. On average, the aluminosilicate compounds in fly ashes from wooden biomass are lower than in coal fly ashes [47]. Similar to the BFA case, their amount can be below 1%. The primary oxide values of CFA are below the range expected for coal fly ash and the BFA has not only very low primary oxides but also high potassium and phosphorus levels (16.19% and 27.68%). Wooden and agricultural biomass ashes are typically enriched in potassium (K), as this is one of the most abundant volatile elements available in raw biomass [48]. In other research papers, the concentration of potassium in wooden biomass has been shown to diminish at temperatures above 900 °C [49]. The temperatures during co-combustion are significantly higher than 900 °C in comparison to biomass combustion, which normally occurs at lower temperatures. In CFA, it is possible that most of the K from the biomass material in this ash volatilized during co-firing, which affects the K concentration in it. K-containing minerals connected with biomass fuel are detected by XRD in CFA in the form of Microcline or Orthoclase and in BFA as Arcanite.

The CFA is more enriched in Mg and Mn oxides compared to RFA. These two elements are essential wooden biomass nutrients described by Dahlquist [50]. BFA is not enriched in these elements because it is primarily composed of phosphate (P), calcium (Ca), and unburned carbon. XRD revealed that portlandite and anhydrite are the only Ca-containing crystalline phase in the BFA and P is built in a crystalline structure of Archerite. Ca content in CFA is also elevated compared to the RFA. XRD analysis detected calcite, portlandite, and a small amount of lime as the Ca-containing crystalline components.

The LOI of the BFA is high for fly ashes (17.98%), which indicates that there is a significant amount of residual volatile matter in that sample. As mentioned, the combustion process for biomass is conducted at a lower temperature than those for the coal and co-combustion fly ashes resulting in a partial firing of the raw biomass. Due to an abundant amount of CaO in the chemical compound of BFA, the signification fraction of the LOI mass loss can be associated with calcium forms of volatile components (CaCO_3_) rather than residual organic carbon [51]. This would also explain why high LOI has not influenced the SSA result of the BFA [52]. CFA with the same coal and similar processing conditions as RFA had higher LOI (7.65%). It can suggest that biomass replacement of coal did have an impact on carbon content for that material. Still, this level of LOI in the EN 450-1 standard qualifies RFA as a fly ash C category.

#### 3.1.2. Particle Size Distributions

Table 2 and Figure 2 show the particles’ diameter characteristics and PSDs for all ashes. The mean value of particle sizes for the coal and co-fired fly ashes were practically similar (consecutively 56.02 and 57.96 μm) in comparison to biomass ash with a 38.47-μm mean diameter. The larger particle size of the RFA and BFA are likely a result [53,54,55] of the incomplete combustion, lower firing temperatures, use of wooden biomass, or all factors mentioned above. From the graphs of particles size distributions (Figure 2), it is possible to assume that the RFA sample has more unified particles’ composition than the other two fly ashes. In CFA, it is possible to observe larger particles with a dimension close to 350 µm. The finest grain composition was observed for BFA where only a few percentages of all particles exceeded 110 µm. For the RFA and CFA used in this study, co-combusting occurs to increase the fineness and, subsequently, reduce the median particle size of the ash compared to the coal fly ash with the same origin. Despite the biomass ash received from agricultural waste seeming to have more fine particles than other fly ashes, this parameter does not consistently increase its fineness.

#### 3.1.3. SEM Analysis

The micrographs on a 10-µm scale clearly revealed particles’ spherical shape for all fly ashes (Figure 3). EDS analysis confirmed that, in RFA and CFA, those particles were mostly built with silica and aluminum compounds typical for glassy aluminosilicates [56]. In BFA, particles were characterized by phosphate, calcium, and potassium compounds present in archerite, which is shown in the BFA structure by XRD.

The RFA ceno-sphere particles were found to be mostly non-porous without any disfigurements (Figure 3a). The EDS analysis in point 47 found vast amounts of ferrous elements (Figure 4), likely associated with hematite observed in an XRD diffractogram (see Figure 1). In CFA, particles with high calcium content were observed. After analysis with Bruker Espirit Spectrum, those particles were defined as lime (Figure 3b and Figure 4c). Figure 3b reveals long, fibrous particles with intact cells of woody morphology. This further indicates that the raw wooden biomass used in CFA production was not fully combusted. Similar observations were noted in other research papers [57]. In BFA, despite lime particles, a vast amount of calcium phosphate irregular plates were built in the morphology structure.

### 3.2. Alkali Activated Mortars

#### 3.2.1. Flexural and Compressive Strengths

Average flexural and compressive strength data for all mortar mixes at 7 and 28 days are presented in Figure 5 and Figure 6. At 7 and 28 days, it was impossible to measure the mechanical properties of RFA-N# samples. It could be related to a slow increase in time of flexural and compressive strengths of these mixes due to insufficient alkaline activation [58,59]. For BFA and CFA mortars with sodium hydroxide (N#) or lime (C#) activators, flexural and compressive strengths increased in most cases between 7 and 28 days. There is a tendency of better strength results for studied mortars in time with the higher molar concentration of N#. An opposite relation was observed for mortars with increased C# addition. Other researchers detected similar relations and they concluded that alkali concentration can be considered as one of the main parameters in the contribution to the mechanical properties of alkali-activated materials. A high level of calcium compounds can affect the geo-polymerization process and cause deterioration of the samples’ microstructure [60,61].

The mortars with mixed activators - NaOH with sodium silicate (N5-S22) and NaOH with CaO (N5-C10)–induced the highest flexural strengths among all combined activated materials. At 7 days, all of the CFA and BFA mixes had flexural strengths greater than 84% of the RFA with the same alkaline activator, excluding BFA-N5-C10 samples. At this age, the RFA and CFA with N5-C10 had the highest strengths. At 28 days, the flexural strength of mortars with C10-S22 remained on the same level, while RFA-N5-S22 mix strength increased primarily from continued geo-polymerization more than twice (Figure 6). The aim of the research was to find the proper alkali activation system for Alkali-activated mortars (AAM) with CFA and BFA in order to have similar technical properties for the RFA-based geopolymer mortars. The most comparable results of flexural strength for mortars based on different fly ashes were obtained in case of the N5-C10 (sodium hydroxide and lime) activation system. Use of N5-S22 (sodium hydroxide and sodium silicate) with CFA and BFA does not allow for the highest value of flexural strength obtained for RFA-N5-S22 (see Figure 6).

The variation in compressive strengths between the alkali-activated ash mortars (Figure 7 and Figure 8) was likely affected by the differing properties of the precursor. Both flexural and compressive strength results can be influenced by particle fineness connected with formation of a higher amount of alkaline alumino-silicate gel, as mentioned in References [62,63,64]. The proper choice of raw material seems more significant for mixes activated with a single alkali-activator. The CFA samples had the highest compressive strengths among other fly ashes activated with N# or C# (the biomass and RFA mortars either did not form a hardened aluminosilicate gel or had low flexural strength results). The increase of molar concentration of NaOH had a similar positive effect on CFA mortars by improving compressive strengths by 22%. In the case of BFA, the compressive strength of BFA-N# mixes was reduced by almost 40% when the activator changed from 5 N to 10 N. At 28 days, compressive strength results of CFA samples with C# were at least 71% higher than the best values of mortars with C# from other fly ashes. Most of the AAM with 15% addition of lime (C15) had compressive strengths below 2.6 MPa.

The differences in strength can be determined by the large amount of active CaO in the raw BFA and CFA. Since the reactive amorphous content of silica-rich and alumina-rich phases has the most influence in geo-polymerization [33,34], the higher glassy content and lower amount of unburned carbon in RFA compared to both CFA and BFA has possibly contributed to its high strength after activation with certain alkali solutions (see Table 2). In consequence, RFA-N5-S22 and RFA-N5-C10 mixes with compressive strengths equal to 14.33 MPa and 10.82 MPa obtained higher results than their equivalents with CFA and BFA. Strength values at 28 days at the other end of the scale had mixes activated with lime and sodium silicate (C10-S22). The compressive strength is below 2.8 MPa. The most promising compressive strength for CFA mortars with a low coefficient of variation (below 5%) was obtained with sodium hydroxide and lime activation (N5-C10). It constituted 84% of the RFA mix with the same alkaline activator. Since the strength results of mortars with single activators were comparatively low, further investigations were conducted only on mortars with mixed alkaline activators.

Obtained results of compressive and flexural strengths for alkali-activated mortars comprise a wide range of results presented by other authors (compressive strength from 2 MPa to more than 120 MPa, flexural strength from below 0.5 MPa to 9 MPa) [65,66,67,68,69,70,71,72]. The lower mechanical properties are either caused by the wrong sand/fly ash ratio [65] or by using only one alkaline activator [66]. In case of a combination of alkaline activators, the enhancement of the mechanical properties is usually obtained through changes in the fly ash characteristics [67], the amount and type of the activator [68,69], or the curing regime [70,71,72].

Hadi et al. [67] analyzed the effects of fly ash characteristics on the compressive strength of geopolymer mortars. His research revealed a correlation between the amorphous SiO_2_ content, the median particle size of the fly ashes, and the compressive strength of tested mortars. Depending on those parameters, analyzed geopolymers presented compressive strength from 7 MPa to 67 MPa after seven days.

De Vargas et al. [68] analyzed compressive strength development of fly ash geopolymer mortars activated by a combined NaOH and Ca(OH)_2_ mixture. After 28 days, samples obtained compressive strength results close to 15 MPa. However, mixes with higher addition of the calcium compound exhibited a reduction in the strength in time.

Gorhan and Kurklu [69], studied the effect of 3, 6, and 9 M sodium hydroxide mixed with sodium silicate solution on the fly ash geopolymer mortar mechanical properties cured at different temperatures up to 24 h. They obtained about 16 MPa and close to 6 MPa in compressive and flexural strength, respectively, after seven days from the mixture made with 6 M NaOH solution cured at a temperature of 65 °C for 5 h.

Based on the literature overview presented above, the mechanical behavior of alkali-activated mortars presented in Figure 5, Figure 6, Figure 7 and Figure 8 could be increased by changing the molar concentration of alkaline activators or by changing the curing conditions. In case of samples with a higher level of a calcium compound, there is a risk of loss in mechanical properties’ results, as in Reference [68].

#### 3.2.2. TGA

The thermogravimetric data for AAM are shown in Table 3 and Figure 9. The bulk weight loss for all mortars was observed below 250 °C, which is related to the evaporation of easily accessible free water (below 105 °C) and water from the gel pores (105 °C–250 °C). Higher weight losses in this region were observed for samples with BFA and mixes with an N5-C10 activator. This can be a result of the higher water absorption of a calcium compound or differences in levels of unburned carbon since carbon particles adsorb water and effectively eliminate it from the pore system by decreasing weight loss connected with free water [73,74].

The high weight loss between 250 °C and 550 °C was observed for all mixes, excluding CFA-N5-C10, which has the most relevant losses above 550 °C associated with carbonated phases. Samples with BFA had mostly a small weight decrease between 410 °C–550 °C likely related to portlandite or soluble phosphates decomposition [75]. The weight loss over 320 °C in all mixes with sodium hydroxide (N5) can be related to disintegration of unreacted NaOH (melting temperature ~ 320 °C). At higher temperatures, distinct weight drops were likely caused by the decomposition of carbonate salts in the pore solution or above an 800 °C release of the unburned carbon from the raw ashes. Weight loss at 800 °C was significant for CFA-C10-S22 and CFA-N5-C10 (4.31% and 5.65%, respectively). In both mixes, we had an additional amount of CaO, which could react with water and CO_2_-forming CaCO_3_. This chemical compound decomposes at a temperature above 550 °C and could be accounted in weight loss in the 550 °C–800 °C area.

The total weight loss at 1000 °C was the highest for mortars with CFA and BFA activated with N5-C10 and C10-S22. However, the most significant loss for them was observed below 550 °C and the weight loss in this area is mostly related to either dehydroxylation of the geopolymer gel with subsequent condensation of the bound silanol or decomposition of Ca(OH)_2_ [76]. The raw CFA, despite high calcium oxide content, has the potential to have aluminosilicate gel through reaction of amorphous silica. In case of the raw BFA rich in phosphate and calcium phases, the weight loss below 550 °C can be only connected with those compounds.

A comparison with the RFA mortars’ TGA data with the CFA showed that weight losses for RFA samples is higher in the CH area for the CFA in the CC sector. When, for CFA-N5-C10, the amount of free and bound water was the lowest among all activated CFA for RFA-N5-C10, those levels were one of the highest. It is possible that, due to this combination of activators, additional aluminosilicate gel products were formed. As a result, this mortar had higher weight loss than with the other activators.

#### 3.2.3. Calorimetry

The temperature evolution profiles of alkali-activated pastes are shown in Figure 10. The fly ashes were mixed with alkaline activators (N5-S22, C10-S22, and N5-C10) in proportions presented in Table 2. Tests were performed during the first 24 h of setting at ambient temperature. Since geo-polymerization is an exothermic reaction [77] and some of the raw materials have a high calcium and phosphorous contents, the results present a combined effect of free calcium oxide hydration, a phosphorous oxide reaction, and a polymerization process.

It can be observed that the temperature increased rapidly in the first minutes after mixing BFA with water. It may be a result of highly exothermic reaction of P_2_O_5_ constituting more than 26% of a BFA chemical compound. The phosphate part soluble in water reacted very quickly with the calcium or sodium oxides, which releases plenty of heat [78]. After achieving its temperature peak, BFA samples show a strong downward trend during the first 5 h and then slowed down, which indicates a significant reduction in the hydration reactions. A similar heat release trend is observed in the hydration of magnesium phosphate cement [79]. All BFA mixes reached high temperature levels with the highest temperature peak for the BFA-N5-C10 at 90.32 °C [80].

#### 3.2.4. SEM

AAM activated with NaOH and CaO (N5-C10) were chosen for SEM analysis as their CFA and BFA activation based on previous conducted tests, which offered similar properties compared to RFA samples. The surface images of the mixes are presented in Figure 11. Figure 11a,c,e display the aggregate (A) and gel (C) interface (B) of alkali-activated mortars. In RFA-N5-C10, the geopolymer gel coated the aggregate without leaving any clear boundary (Figure 11a). It is possible that either a chemical reaction between the aggregate and alkali environment occurred or a mechanical interlocking occurred between both phases. This strong chemical bonding between the geopolymer gel and aggregate may be a reason for the higher flexural and compressive strengths of the RFA mortar [71,81]. Similar coating of aggregate in CFA-N5-C10 microcracks (Figure 11c, area D) have been noted. It may be related to a higher temperature during setting since more extensive cracks were also seen in BFA-N5-C10 SEM images (Figure 11d). Between the gel phase and aggregate in BFA-N5-C10, a visible interfacial transition zone was observed. It may signify poor adhesion between both phases, which is followed by lower mechanical properties.

A typical fly ash composite mainly consists of aluminosilicate gel or geopolymer paste as well as unreacted particles of fly ash and voids [81,82]. The densest microstructure is observed in the RFA-N5-C10 sample with visible aluminosilicate gel (Figure 11b, point 29). Despite higher reactivity, the gel has an irregular shape and there still exist unreacted, spherical fly ash particles. The higher porosity and coarser microstructure of the gel with some amount of unreacted or partially reacted particles in the CFA-N5-C10 mortar (Figure 11d, points 38, 39) indicates a moderate geopolymer reactivity in this sample. The EDS analysis of the chemical compound in point 39 revealed calcium and silicon, which may be a part of the C-S-H gel responsible for mechanical properties in this material.

The BFA-N5-C10 mix has a significantly different morphology when compared to RFA and CFA mixes. The porous, coarse gel structure is similar to particle frameworks of amorphous calcium phosphate [83]. This similarity was confirmed by the EDS analysis in points 84 and 85 (Figure 11f) characterized by the main elements: phosphorus, calcium, and oxide. The spherical unreacted particle embedded in this gel (point 83, Figure 11f) was identified as archerite, as shown by XRD. The BFA-N5-C10 sample had no visible aluminosilicate structures in SEM analysis.

Compared to other mortars, RFA-N5-C10 formed a more visually homogenous gel structure with a denser, less porous matrix and fewer partially reacted particles. Nonetheless, CFA-N5-C10 had a mostly uniform microstructure whereas the microstructure of BFA-N5-C10 were porous and abundant in phosphate compounds.

## 4. Conclusions

The influence of the activator type on the formulation of modified fly ash based geopolymer mortars was studied on different fly ashes. Based on the results, key findings can be drawn.

Mortars activated with NaOH and Sodium Silicate (N5-S22) and NaOH with CaO (N5-C10) induced the highest flexural and compressive strengths for all the AAM. At 28 days, compressive strength results of CFA mortars with CaO addition were at least 71% higher than for other samples,The increase of NaOH molar concentration had a positive effect on mechanical properties of CFA mortars by improving its flexural and compressive strength results by 15% and 22%, respectively,Compressive strengths of geopolymer mortars RFA-N5-S22 and RFA-N5-C10 with equal 14.3 MPa and 10.8 MPa after 28 days were the highest of all mixes. The higher glassy content, a lower amount of unburned carbon in RFA with a more visually homogenous gel structure as well as a denser, less porous matrix of RFA-based AAM when compared to CFA, BFA, and their mortars, which have possibly contributed to its high strength after activation,Using the activator had a significant influence on the maximum temperature of the alkali-activated pastes setting. The N5-C10 in comparison to the N5-S22 activator increased the setting temperature by more than 45% for CFA and almost 36% for RFA. The activators from the most to the least effective on the setting temperature are arranged as: N5-C10 > C10-S22 >N5-S22,The Biomass Fly Ash (BFA) due to low alumina and silica content is not adequate for alkali-activation. The main chemical structures were connected with hydration of active CaO and creation of CaCO_3_. Solidification in an acidic environment may be analyzed due to a high level of P_2_O_5_.

Further research has to be performed with regard to durability and robustness of the mixes in several aggressive environments.

## Figures and Tables

**Figure 1 materials-13-01033-f001:**
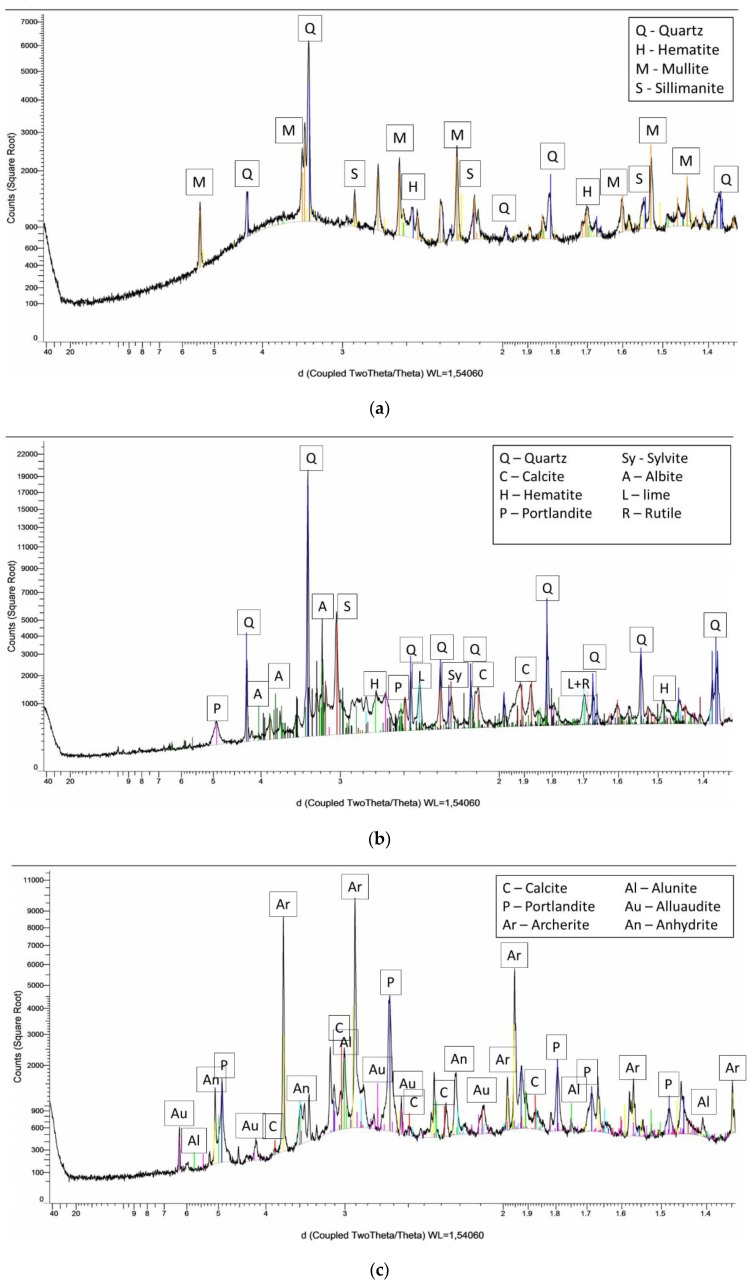
The X-ray diffraction test (XRD) diffractograms: (**a**) silica fly ash from coal combustion (RFA), (**b**) co-combustion fly ash (CFA), (**c**) biomass fly ash (BFA).

**Figure 2 materials-13-01033-f002:**
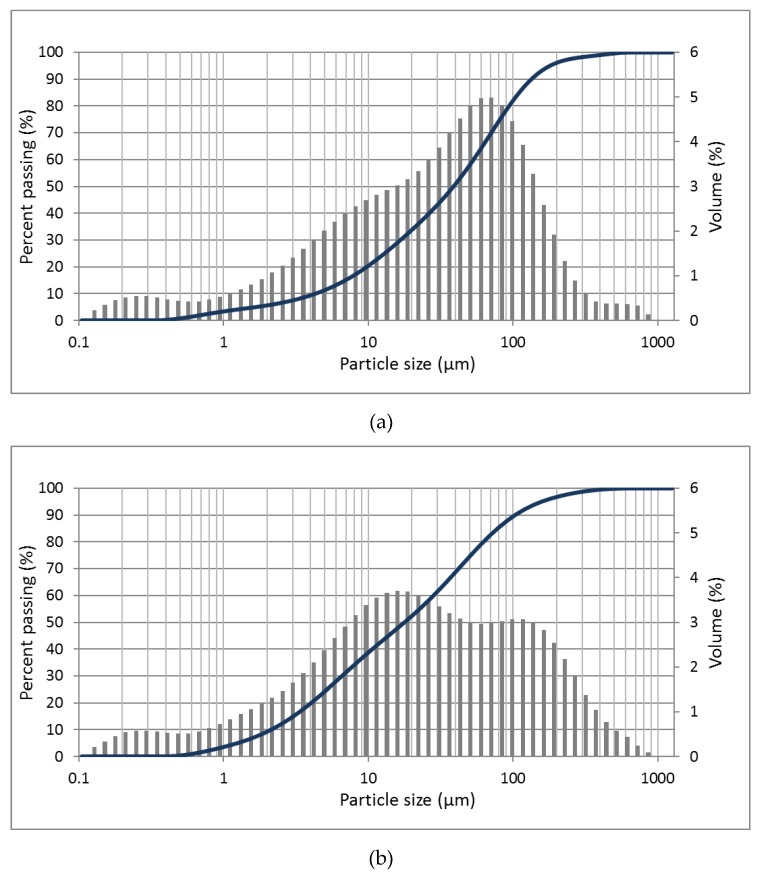
Raw materials’ particle size distribution (**a**) RFA, (**b**) CFA, and (**c**) BFA.

**Figure 3 materials-13-01033-f003:**
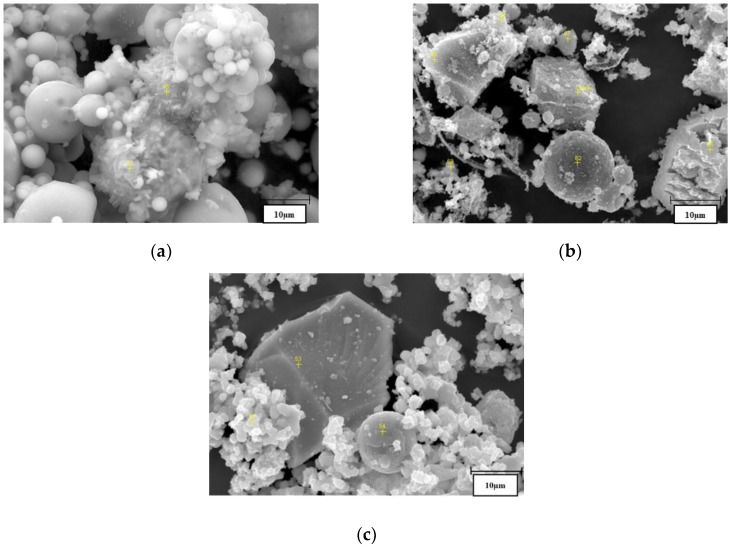
SEM pictures in 10-µm scale of fly ashes: (**a**) RFA, (**b**) CFA, and (**c**) BFA.

**Figure 4 materials-13-01033-f004:**
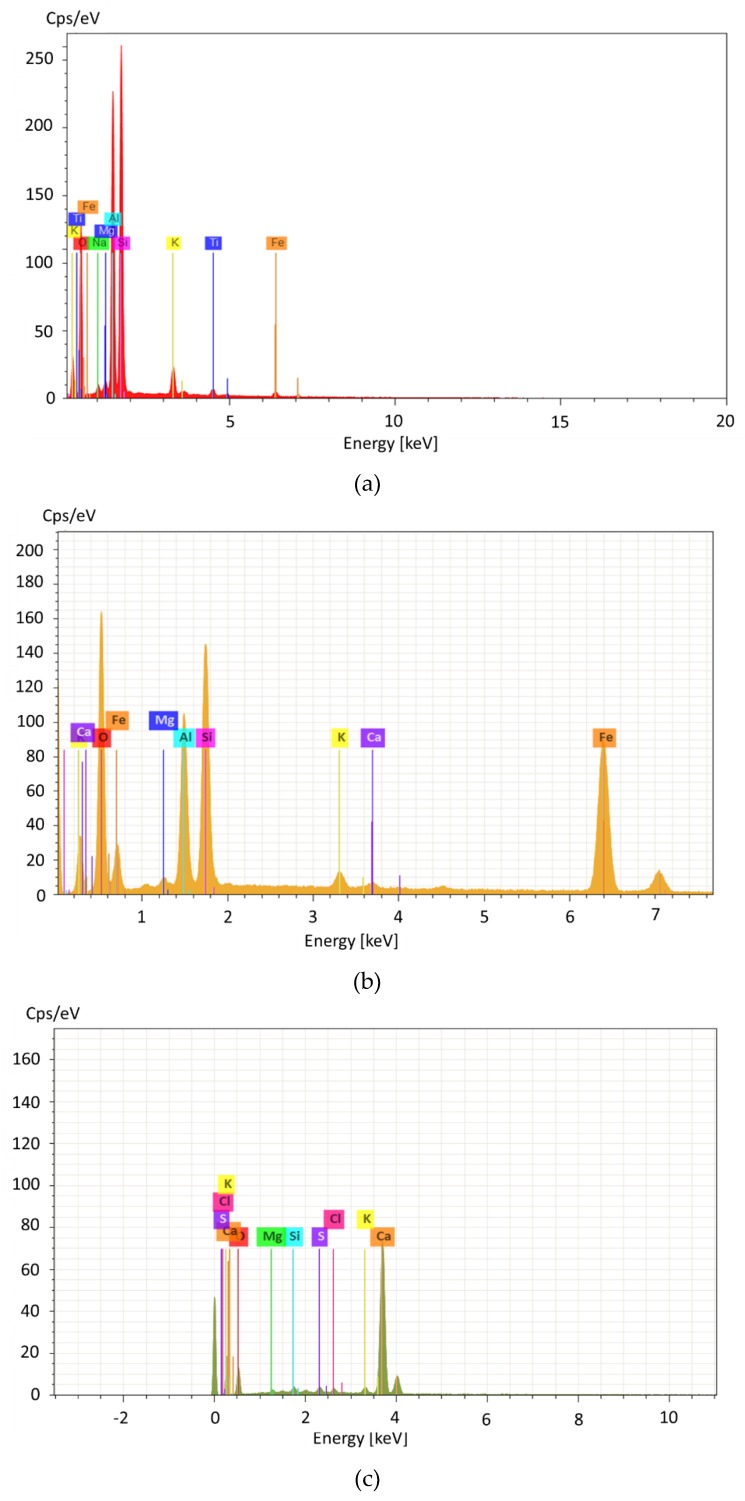
Examples of Espirit Spectrum Analysis for Bruker EDS: (**a**) RFA, point 42, aluminosilicate particles, (**b**) RFA, point analyzed as mullite, and (**c**) CFA, point analyzed as lime.

**Figure 5 materials-13-01033-f005:**
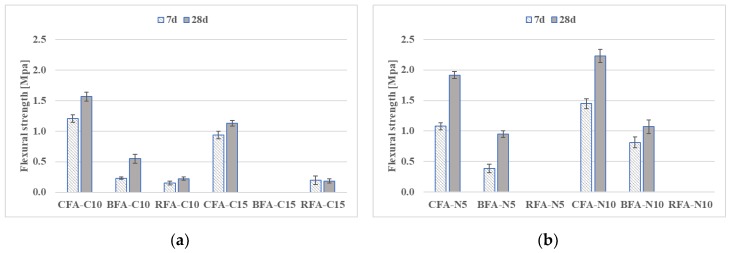
Flexural strength development of mortars with several activators: (**a**) 10% and 15% addition of CaO, and (**b**) 5 M and 10 M of NaOH.

**Figure 6 materials-13-01033-f006:**
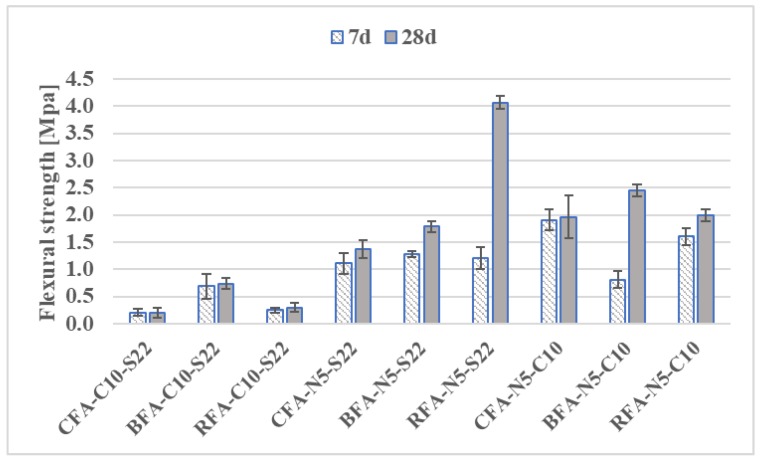
Flexural strength development of mortars with mixed activators.

**Figure 7 materials-13-01033-f007:**
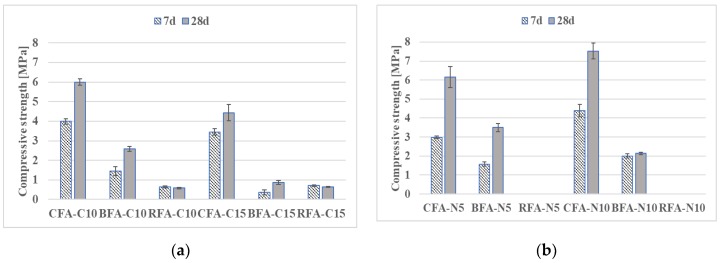
Compressive strength development of mortars with various activators: (**a**) 10% and 15% addition of CaO and (**b**) 5 M and 10 M of NaOH.

**Figure 8 materials-13-01033-f008:**
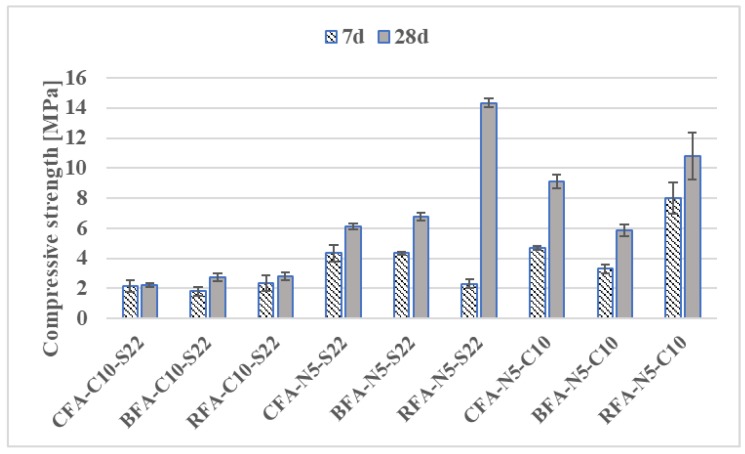
Compressive strength development of mortars with mixed activators.

**Figure 9 materials-13-01033-f009:**
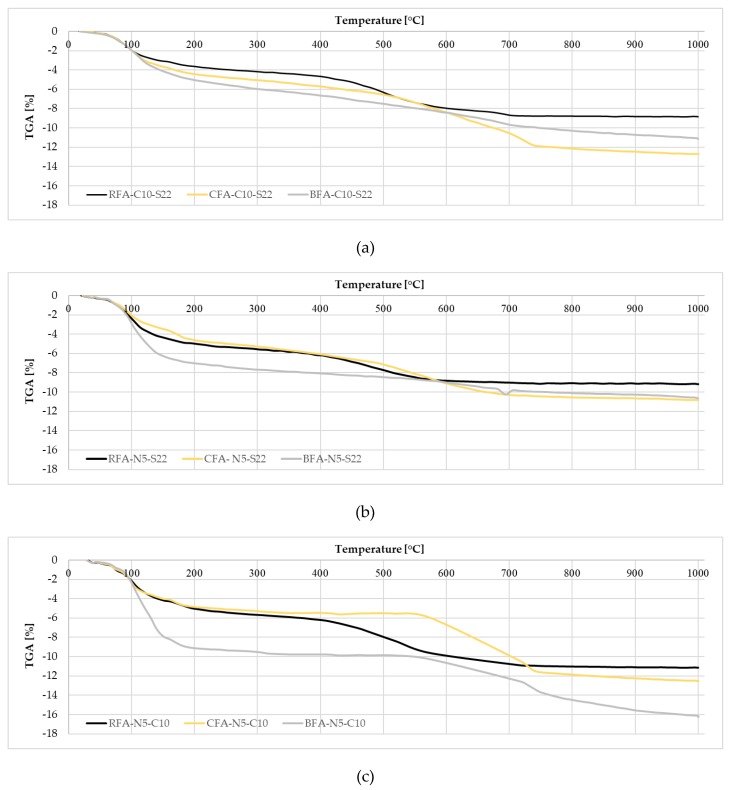
Thermogravimetric analysis (TGA) curves in the 22–1000 °C temperature range for mortars activated with: (**a**) C10-S22, (**b**) N5-S22, and (**c**) N5-C10.

**Figure 10 materials-13-01033-f010:**
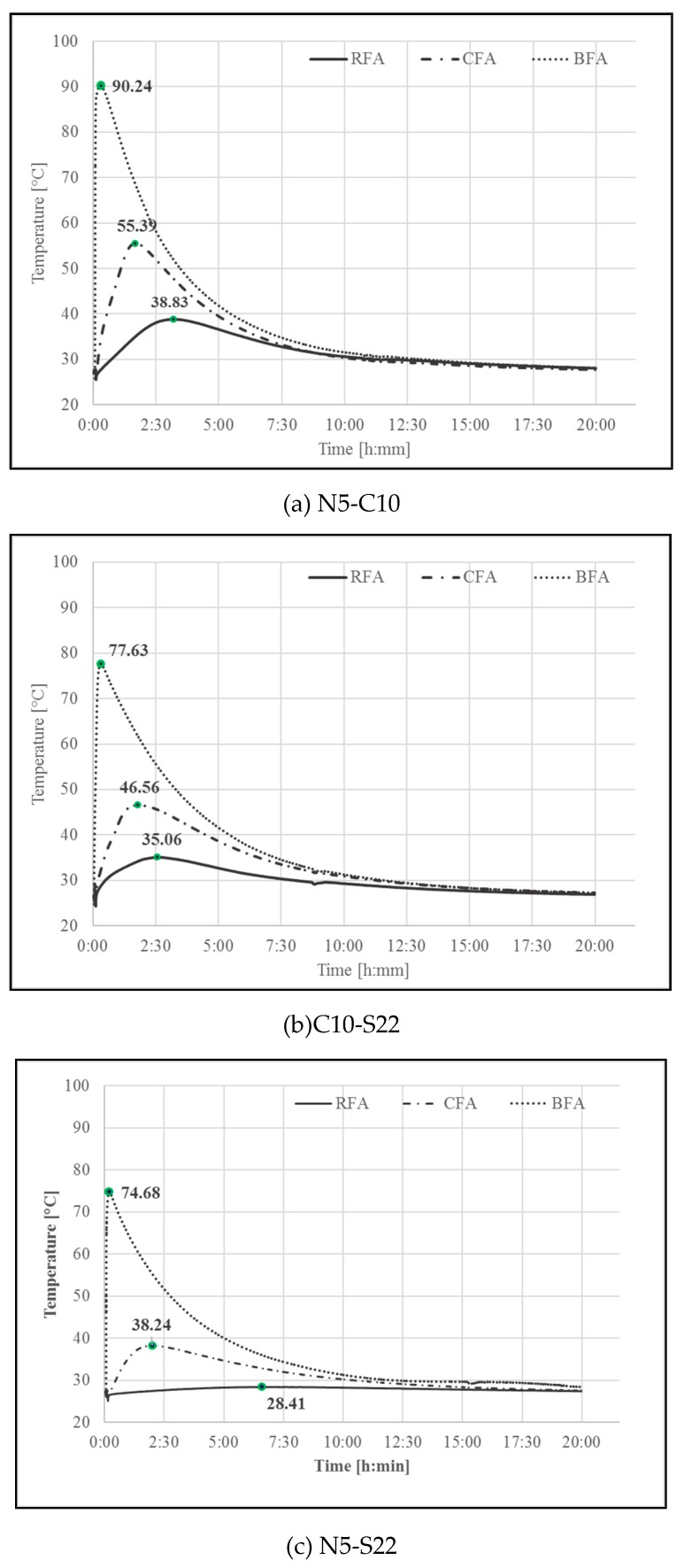
Temperature profiles of pastes with different activators (**a**) N5-C10, (**b**) C10-S22, and (**c**) N5-S22.

**Figure 11 materials-13-01033-f011:**
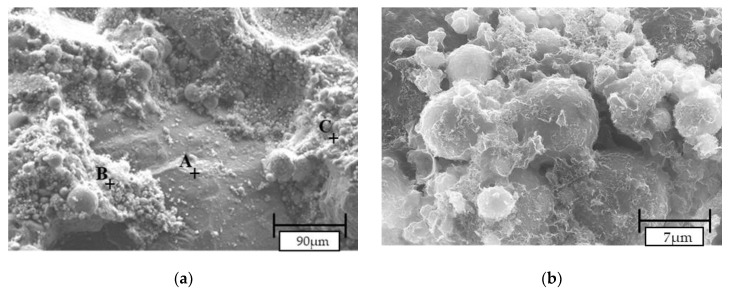
SEM pictures of mortars alkali-activated with N5-C10: (**a**,**b**) RFA, CFA (**c**,**d**), and BFA (**e**,**f**). * A—aggregate, B—interface between gel and aggregate, C—aluminosilicate/geopolymer paste, and D—microcracks.

**Table 1 materials-13-01033-t001:** Mortar mix designs.

Mix Type	Raw Material (g)	Sand (g)	NaOH Solution (g)	Silicate Solution (g)	CaO (g)	Water (g)
RFA-N5	450	1350	225	0	0	0
CFA-N5	450	1350	225	0	0	0
BFA-N5	450	1350	225	0	0	0
RFA-N10	450	1350	225	0	0	0
CFA-N10	450	1350	225	0	0	0
BFA-N10	450	1350	225	0	0	0
RFA-C10	405	1350	0	0	45	225
CFA-C10	405	1350	0	0	45	225
BFA-C10	405	1350	0	0	45	225
RFA-C15	382	1350	0	0	68	225
CFA-C15	382	1350	0	0	68	225
BFA-C15	382	1350	0	0	68	225
RFA-C10-S22	405	1350	0	100	45	225
CFA-C10-S22	405	1350	0	100	45	225
BFA-C10-S22	405	1350	0	100	45	225
RFA-N5-S22	450	1350	225	100	0	0
CFA-N5-S22	450	1350	225	100	0	0
BFA-N5-S22	450	1350	225	100	0	0
RFA-N5-C10	405	1350	225	0	45	0
CFA-N5-C10	405	1350	225	0	45	0
BFA-N5-C10	405	1350	225	0	45	0

**Table 2 materials-13-01033-t002:** Raw material characteristics.

Chemical Composition (%) and Physical Characteristic	RFA	CFA	BFA
SiO_2_	50.56	46.57	0.45
Al_2_O_3_	26.20	4.15	0.12
Fe_2_O_3_	6.21	3.40	0.22
MnO	0.06	0.66	0.02
MgO	2.63	3.27	1.19
CaO	2.70	21.90	30.95
Na_2_O	0.94	0.98	1.20
K_2_O	3.51	5.58	16.19
TiO_2_	1.13	0.21	0.00
P_2_O_5_	0.55	3.01	27.68
Loss On Ignition, LOI (%)	5.51	7.65	17.98
Total of XRF	100.01	97.37	96.00
∑ (SiO_2_ + Al_2_O_3_ + Fe_2_O_3_)	82.97	54.12	0.79
Real density (g/cm^3^)	2.188	2.732	2.453
Mean particle size (μm)	56.02	57.96	38.47
BET SSA * (m^2^/g)	3.4489	5.5573	2.0971
**Mineral Composition (%)**			
Quartz	7.02	18.72	–
Mullite	14.63	–	–
Calcite	–	10.6	–
Portlandite	–	3.45	15.51
Microcline	–	3.33	–
Orthoclase	–	3.18	–
Microcline	–	–	–
Alunite	–	–	2.4
Anhydrite	–	–	2.41
Arcanite	–	–	10.96
Alite	–	–	–
Archerite	–	–	23.61
Amor	76.91	53.13	41.35

* Brunauer–Emmett–Teller specific surface area

**Table 3 materials-13-01033-t003:** The free water, the bound water of the hydrate phases, the Ca(OH)_2_, and the CaCO_3_ content in mortars.

Mix	Free H_2_O, %	H *, %w	CH *, %w	CC *, %w	Total
	<105 °C	105 °C–550 °C	410 °C–550 °C	550 °C–800 °C	%w
CFA-C10-S22	2.33	5.56	1.70	4.31	12.70
BFA-C10-S22	2.30	5.96	1.25	2.13	11.15
RFA-C10-S22	2.23	5.33	2.58	1.21	8.84
CFA-N5-S22	2.39	5.92	1.90	2.30	10.88
BFA-N5-S22	3.57	5.51	0.71	1.15	10.66
RFA-N5-S22	2.77	5.75	2.12	0.59	9.18
CFA-N5-C10	3.11	3.21	0.47	5.65	12.51
BFA-N5-C10	3.81	7.01	0.48	3.96	16.26
RFA-N5-C10	3.06	6.70	3.21	1.29	11.16

* H—Bound H_2_O, CH—Calcium Hydroxide (Ca(OH)_2_, CC—Calcium Carbonate(CaCO_3_).

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
