# Peer review of "Influence of Activators on Mechanical Properties of Modified Fly Ash Based Geopolymer Mortars"

_materials, 2020, doi:10.3390/ma13051033_

Round 1
Reviewer 1 Report
The paper “Influence of activators on mechanical properties of modified fly ash based geopolymer mortars” is an interesting article regarding the properties of geopolymers deriving from different types of fly ashes. The topic is current and of quite large interest, but in my opinion the overall quality of the manuscript is not high. The most relevant drawback of the manuscript deals with its very informative character: a very large amount of data is presented and scholarly described, but no discussion of the results can be recognized. It is for example not clear the role of x-ray diffraction, which is mentioned in the experimental part as one of the employed techniques: do data contained in Tables 1 and 2 derive from XRD? Another issue regards the Introduction, which can be clearly understood only by people already familiar with the specific research field. Just few lines describing a geopolymer, as well as the role of fly ashes could be of help for the reader to focus on the topic. My advice for Authors is to try to better harmonize all the presented data within an exhaustive discussion.
Other issues are the following ones:
Please define the FA acronym. It is advisable to define the loss on ignition and what the corresponding number refers to. Figure 1. Data related to the raw materials particle size distribution are presented quite in a strange way. It is preferable to show a (lognormal?) distribution as size distribution vs. size. This way it is easier to recognize the mean particles size. Figure 2. What does “in 10 micron scale” mean? I suggest to add a scale marker within microphotographs. Figure 2. How was the “glassy” feature of RFA and CFA particles recognized? Figure 2 and Figure 8. Please eliminate numbers within microphotographs. Figure 3 and Figure 5. Panels a) and b) are most probably exchanged. Line 250. What do Authors mean with “disintegration of the activator?” Authors should report instead of (or in addition to) the scarcely readable Table 3, at least a thermogram directly showing the described weight losses.
Reviewer 2 Report
Due to the environmental concerns associated with cement production, the behavior of alternative additives to cement has received attention. In this paper, research was performed to investigate the effect of various activators on fly ash-based geopolymer mortar. However, the paper is not well structured and prepared for publication. The paper must be significantly improved and the number of errors and typos must be minimized.
Abstract: line 16-21 is too long and confusing. Please revise it.
Since there are a number of similar research works in the field, what is the novelty of this research paper and how the data can add value to the literature. Acronyms must be carefully defined. For example ordinary Portland cement in abstract Lines 35, 41 and 45: CO2 not CO2 Line 54: what is FA: fly ash? So it needs to be defined when it shows up in the text for the first time XRF, XRD, TGA and SEM procedure must be explained Materials and method section needs a major revision to clearly demonstrate mix proportion, tests, and etc. I suggest to include a few subheadings and move the relevant materials under each of those subheadings. “The chemical composition and physical characteristic of the fly ashes is presented in Table 1”. Do you mean Table 2? What approach was used to quantify the composition of fly ashes? It should be explained. The authors stated they used XRD for characterization but they did not show the spectra. Lin e139: “XRD analysis detected calcite, portlandite and small amount of lime as the Ca-containing crystalline components.” Did author perform XRD? Why such important data are missing (spectra)? Lin e150: “Table 1 and Figure 1 show the particles…” Table 1 shows different thing than particle size Line 163: “The optical micrographs …” the authors performed scanning electron microscope not optical one. Line 166: again the results of XRD are missing. Line 164: EDS analysis: The authors need to include EDS results to support the claim. 2: As written in the image, how the authors concluded the presence of CaO in Fig. 2b? SEM/EDS can only detect the elements not a compound. Line 171: what is Table 1? Line 170: “The surface of RFA particles was found to be compact” this is not a correct statement as the sampling can affect how the powder looks like in SEM Line 179-180: Figure 3: Please check the y-axis values. For example: 2.5 MPa or 2,5 MPa? Line 103: table 1 or 2? There are many typos and errors in the manuscript. Line 103: please include the corresponding symbol for activator too (I meant C, N and C-S) Figure 3 is incomplete (x-axis) Figure 3: its caption seems that Fig.3 a is 5Man d10M NaOH while Fig 3 b is CaO. Why no strength development was observed for some samples in Fig 2 and 3 from 7days to 28 days? Lin e249: Please check for typo “320°C In…” Lin e308: please check the citations. Lin e319: what were the methods of XRD preparation and working conditions? Please include the XRD and EDS spectra to support the statement For the calorimetry section, I suggest relying on other(s) reviewers’ comment(s) as I am not an expert in that field.Author Response
Please see the attachment.

Reviewer 3 Report
The authors investigated the influence of the type of activator on the formulation of modified fly ash based geopolymer mortars.
In Introduction, it is suggested to provide more detailed overview of the conducted experimental research. It is extremely imported to highlight the novelty of the provided research and experiments. Also, it should be stated what is scientific contribution of the paper.
In order to make comparison of provided results it is suggested to add existed results from similar research and to provide discussion of the obtained results. The discussion of the results should be added with critically explanations of advantages and disadvantages in comparison with similar materials and the obtained results.
Reviewer 4 Report
The study carried out is interesting but its novelty is not clear; authors must emphasize the novelty of the study. Moreover, in some parts of the manuscript there is a lack of discussion and comparison of the results with the literature reports. Other specific comments are:
Introduction: write CO2 instead of CO2.Introduction: the aim of the study is not clearly exposed in the introduction. The manuscript is supposed to deal “the influence of activators” and the activators are not mentioned in the introduction.
Materials and methods: describe the tests made in the FAs; also describe the tests carried out in the mortars (Flexural and compression tests, Thermogravimetric analysis (TGA), Scanning Electron Microscopy 107 (SEM) and Calorimetry).
Results and discussion, line 122: How did you measure the amorphous content? Moreover, 41,35% is the amorphous content of BFA, not the one of CFA, according to Table 2.
Results and discussion, compressive strength results: How do you explain the very low compressive strength values obtained in all the cases compared to previous results of FA bases geopolymer mortars? The comparison of your results with other results already published is very advisable.
Results and discussion, lines 227-230: Why do you extract this conclusion?
Results and discussion, lines 241: It is weird call “free water” the water of the gels evaporable from 105ºC to 250ºC…
Round 2
Reviewer 1 Report
The manuscript was extensively revised, and I think that it can now been accepted for publication. Just some minor issues remain, such as markers in Figure 11, which should be better placed.
Author Response
Dear Sir/ Madam
On behalf of myself and other authors, thank you for your review of the article.
The markers corrections were applied. We hope that now the paper meets the expectations.
Best regards,
Authors
Reviewer 2 Report
All comments have been carefully addressed and at this stage, I support the manuscript for publication.
Author Response
Dear Sir/Madam
On behalf of myself and other authors, thank you for your review of the article.
Best regards,
Authors
Reviewer 4 Report
Authors have improved the manuscript but there is still a lack of discussion of the results obtained in the alkali activated mortars (section 3.2). This section has not been modified at all and without an adequate discussion of the obtained results the manuscript cannot be published.
Author Response
Dear Sir/Madam,
On behalf of myself and other authors, thank you for your review of the article. The suggestion indicated in the review was applied. We hope that now the paper meets the expectations. In case of necessity of making further corrections, I am looking forward to hearing from you.
Best regards,
Authors
1.This section (section 3.2) has not been modified at all and without an adequate discussion of the obtained results the manuscript cannot be published
New paragraph has been added starting from line 377.